# Outcomes of Vitrectomy with Fovea-Sparing and Inverted ILM Flap Technique for Myopic Foveoschisis

**DOI:** 10.3390/jcm11051274

**Published:** 2022-02-25

**Authors:** Yu Wakatsuki, Hiroyuki Nakashizuka, Koji Tanaka, Ryusaburo Mori, Hiroyuki Shimada

**Affiliations:** Department of Ophthalmology, Nihon University School of Medicine, Tokyo 101-8309, Japan; nkshizuk@gmail.com (H.N.); tanaka.koji@nihon-u.ac.jp (K.T.); mori.ryusaburo@nihon-u.ac.jp (R.M.); shimada.hiroyuki@nihon-u.ac.jp (H.S.)

**Keywords:** myopic foveoschisis, fovea-sparing ILM, inverted ILM flap, vitreous surgery

## Abstract

Surgical treatment of myopic foveoschisis (MF) can result in a macular hole in 11–17% of patients that may lead to poor visual outcomes and progression to macular hole retinal detachment. We evaluated the benefit of vitrectomy to treat MF using the inverted internal limiting membrane (ILM) flap and fovea-sparing ILM techniques. We studied 20 eyes of 20 patients (7 men, 13 women) with high MF (mean axial length, 29.3 ± 1.7 mm). MF was classified by optical coherence tomography findings: retinoschisis (7 eyes) or foveal detachment (13 eyes). Between October 2013 and June 2021, we performed vitreous surgery in all 20 patients, employing both techniques. Air tamponade was used in 4 eyes, SF6 gas in 10 eyes, and C3F8 gas in 6 eyes. All patients stayed in the face-down position for one full day postoperatively. Visual acuity and foveal contour were analyzed using optical coherence tomography before surgery and at 3 and 6 months postoperatively. LogMAR visual acuity was 0.46 before surgery, with a significant improvement at 3 months (0.34) and at 6 months (0.2) postoperatively (*p* = 0.024, *p* < 0.001, respectively). In all patients, the foveal contour showed improvement without macular hole formation after surgery. These results show that vitrectomy, performed using the inverted ILM flap and fovea-sparing ILM technique, is effective for treating MF.

## 1. Introduction

Myopia is on the rise in Japan and other Asian countries and constitutes a common disease in daily clinical practice. If the myopia leads to pathologic myopia, visual impairment may result through posterior staphyloma, chorioretinal atrophy and choroidal neovascularization [1]. Myopic foveoschisis (MF) was first reported in 1958 as macular retinal detachment without a macular hole [2]. Subsequently, the term MF was defined for the first time by Takano et al. in 1999, using optical coherence tomography (OCT) as foveal retinoschisis and retinal detachment associated with high myopia [3]. MF is observed in 9–34% of eyes with high myopia or posterior staphyloma, defined by a refractive error less than −6.0 D and/or an axial length greater than 26 mm [4,5,6,7]. MF is often asymptomatic initially. However, complications such as foveal retinal detachment, macular hole and macular hole retinal detachment can lead to visual impairment, requiring vitreous surgery when visual acuity deteriorates.

Vitreous surgery combined with internal limiting membrane (ILM) peeling has been reported to be beneficial for MF [8,9,10]. However, postoperative macular hole formation has been reported in 7.5–21.4% of MF cases without an initial macular hole [11,12,13,14,15,16]. To reduce this risk, Shimada et al. reported a new surgical technique that preserves the ILM of the fovea [16]. This fovea-sparing ILM peeling to treat foveal retinal detachment with MF has improved visual function and anatomy.

MF can progress to a macular hole or macular hole retinal detachment in 2–3 years in about 50% of patients [17,18]. In addition, postoperative macular holes associated with high myopia have a lower closure rate [19,20,21]. Michalewska et al. reported that the rate of macular hole closure could be increased by inverting the ILM flap for operations involving large macular holes [22]. Recently, a modification of the Inverted ILM technique has been reported, in which only a portion of the ILM is debrided to reduce the debrided area and minimize surgical trauma. Decreasing the area of ILM peeling has the advantage of inducing fewer changes in the central retina [23,24]. Kuriyama et al. reported the effective use of the inverted ILM technique for macular holes in highly myopic eyes, including those with and without retinal detachment [25]. These new surgical methods may be effective in the treatment of MF with macular holes.

Using OCT, Ikuno et al. classified MF into 3 types: macular hole, foveal detachment and retinoschisis [10]. This OCT classification is considered useful for determining surgical indications. The fovea-sparing ILM method is useful for the foveal detachment and retinoschisis types classified by Ikuno et al. [16]. However, in high myopia, it is sometimes difficult to differentiate macular holes from foveal detachment and retinoschisis by OCT alone because of derogation of OCT due to high myopia. In addition, a macular hole might be created during surgery when we perform epimacular membrane removal and ILM peeling for foveal detachment and retinoschisis.

In this report, we devised a new technique to make the treatment of MF without macular holes safer through a combination of fovea-sparing and inverted ILM flap techniques.

## 2. Materials and Methods

This report includes 20 patients (7 men and 13 women) who underwent vitrectomy for MF without macular holes between October 2013 and June 2021. Their mean age was 62.1 ± 11.1 years, mean axial length 29.3 ± 1.7 mm, and mean follow-up observation period 20.9 ± 22.0 months. The inverted ILM flap method combined with a fovea-sparing technique was performed in all patients, and they all stayed in the face-down position for one full day postoperatively. LogMAR visual acuity, foveal contour and ellipsoid zone (EZ) continuity were observed before surgery, 3 and 6 months after surgery, and at the last follow-up. The morphology of the fovea shown by OCT was classified as retinoschisis or foveal detachment (Figure 1).

The disappearance of foveal detachment or retinoschisis was defined as “recovered”; insufficient disappearance was defined as “improvement”; persistence without any change was defined as “unchanged”; and deterioration of the foveal contour was defined as “worse”. EZ continuity was examined in the horizontal section of OCT and classified as “absent”, “irregular” or “continuous”. The confirmation of foveal contour and EZ continuity as shown by OCT was performed by 2 examiners (Y.W. and H.N.).

All patients provided informed consent, and we performed this study which complied with the guidelines of the Declaration of Helsinki with approval from our institutional review board.

### 2.1. Surgical Methods (ILM Flap Procedure)

All surgeries were performed by a single experienced surgeon (N.H.). Phacoemulsification with intraocular lens implantation was performed in phakic eyes, followed by vitreous surgery. A 25- or 27-gauge standard 3-port pars plana vitrectomy using a constellation vision system (Alcon Laboratories, Inc., Fort Worth, TX, USA) and a wide-angle non-contact viewing system (Resight; Carl Zeiss Meditec AG, Yena, Germany) was performed under local anesthesia (retrobulbar anesthesia: 2% lidocaine). Three ports were placed in the infratemporal, supratemporal and supranasal quadrants 3.5 mm posterior to the limbus, and a core vitrectomy was performed using triamcinolone acetonide (Macuaide^®^, Wakamoto Pharmaceutical Co., Ltd., Tokyo, Japan). A posterior vitreous detachment was created if it did not exist. If the vitreous cortex remained at the posterior pole, it was removed using diamond dust or a back-flush needle. After peripheral vitreous shaving using scleral compression, the ILM was stained with 0.0625% Brilliant blue G (Sigma Aldrich, St. Louis, MO, USA). Firstly, we peeled the ILM where the retinoschisis existed, except in the foveal area (about 1.5 disc area) and the superior foveal area. Secondly, we inverted the ILM flap of the superior foveal area to the fovea where the ILM was intact. Finally, we performed fluid-air exchange as gently as we could to keep the inverted ILM stable on the fovea. In particular, we mainly used the stream of air-fluid exchange to slightly tilt the eyeball downward and the fluid-air exchange via the flap in front in order to turn the flap (Figure 2).

### 2.2. Statistics

Statistical analyses were performed using IBM SPSS Statistics, version 28.0 (IBM Corp., Armonk, NY, USA). A Wilcoxon signed-rank test was used to compare visual acuity before surgery, with 3 and 6 months after surgery. A *p*-value of <0.05 was taken to indicate a significant difference.

## 3. Results

A 25-gauge vitrectomy was performed in 12 patients and a 27-gauge vitrectomy in 8 patients. Fourteen patients underwent combined surgery (cataract and vitreous surgery) and 6 patients underwent vitreous surgery alone. Seven eyes had retinoschisis and 13 eyes had foveal detachment. Air tamponade was used in 4 eyes, SF6 gas in 10 eyes, and C3F8 gas in 6 eyes (Table 1).

LogMAR visual acuity was 0.46 ± 0.22 preoperatively, 0.34 ± 0.25 at 3 months postoperatively, and 0.20 ± 0.18 at 6 months postoperatively, showing significant improvement at 3 months and 6 months postoperatively compared with preoperatively (*p* = 0.024, *p* = 0.00013, respectively; Wilcoxon signed-rank test; Figure 3).

At the last follow-up observation, foveal contour was improved in all patients without forming a macular hole. In addition, EZ continuity was present in 2 eyes at 3 months after surgery, 10 eyes at 6 months after surgery, and 14 eyes at the last follow-up observation (Table 2 and Figure 4).

No postoperative complications, including postoperative macular hole formation, were observed (Figure 5). In Patient #5, who had retinoschisis before surgery, an outer lamellar macular hole (OLMH) was observed after surgery (Patient 5 in Table 1). However, the OLMH and retinoschisis disappeared and the foveal contour was improved at the last observation, 1 year after surgery, although EZ discontinuity remained (Figure 6).

## 4. Discussion

In this study, we performed vitrectomy combined with a fovea-sparing and inverted ILM flap technique for MF without macular holes. The foveal contour was improved in all patients without postoperative macular hole formation during the observation period, which ranged from 6 months to a maximum of 93 months.

When a macular hole is accompanied by MF, meta-analyses suggest that the inverted ILM peeling flap technique has a better macular hole closure rate than complete ILM peeling [26]. On the other hand, ILM peeling at the fovea might increase the risk of postoperative macular hole formation after vitreous surgery for MF without macular holes [10,11,12,13,14]. This is because the retinal tissue at the fovea is fragile, especially in myopic eyes, and Müller cell cone detachment might occur during ILM peeling resulting in macular hole formation. Therefore, for MF without macular holes, a fovea-sparing ILM method, which preserves the ILM at the fovea, is helpful in preventing postoperative macular hole formation [16,27,28].

Furthermore, in patients with high myopia, there may be a potential macular hole that preoperative OCT cannot identify, or a macular hole might be created during surgery due to posterior vitreous detachment and/or removal of the vitreous cortex [29]. Additionally, foveal detachment in MF may progress to a macular hole or macular hole retinal detachment [18]. Therefore, the concern of postoperative macular hole formation still remains even if we perform a fovea-sparing method. Indeed, Tian et al. reported that full-thickness macular hole formation was observed in eyes that underwent fovea-sparing ILM peeling, and in these patients, an OLMH was detected preoperatively [30].

To reduce the risk of postoperative macular hole formation, we devised a surgical method combining fovea-sparing and inverted ILM peeling and performed this new technique to treat retinoschisis and foveal detachment types of MF without macular holes. By peeling the ILM except at the fovea, we were able to cover the fovea by the doubled ILM, which might prevent the formation of a postoperative macular hole more safely. At the same time, we believe that this technique is more helpful than fovea-sparing ILM alone, as unnoticed tiny macular holes may not be detected by preoperative OCT or intraoperative observation, and the addition of the inverted ILM flap method allows us to deal with these small macular holes or those that may form intraoperatively. Although our study did not evaluate other techniques, there was no postoperative macular hole formation using our combination method of inverted ILM flap and fovea-sparing ILM technique for MF. Very recently, a similar surgical technique to ours was reported for MF [31]. Lin et al. retrospectively compared the combination of an inverted ILM flap and fovea-sparing ILM technique for MF. The incidence of postoperative macular holes was 9.7% with the fovea-sparing ILM technique which was significantly different to the 0% incidence with the combined ILM flap and fovea-sparing ILM techniques. Furthermore, Lin et al. said the ILM fragments could be detected by OCT in some layers on the foveal inner surface, but they have not followed the postoperative course of retinal structures such as EZ.

On the other hand, we observed the morphology of the EZ in all cases after surgery, and some post-operative OCT showed fragments of the inverted ILM flap in the superficial layer of the macular retina (Figure 7). There was a concern that this might interfere with EZ recovery or improvement of foveal contour, but we observed the morphology of the EZ in all eyes after surgery, and at last observation, EZ improvement was observed in 70% of eyes, indicating that the ILM flap does not interfere with regeneration of the EZ.

Shimada et al. reported focal retinoschisis was observed at the site of retinal vascular microfolds after surgeries using both the fovea-sparing technique and the complete ILM-peeling technique [15]. Ho et al. reported the occurrence of postoperative parafoveal elevation in the area without ILM peeling in his study, which might be caused by ILM traction at the peeling margins. They also concluded that the ILM preserved during surgery should be as small as the size of the foveola to release all the tangential traction, because some foveoschisis involves the very center of the fovea. [32]. In our method, the parafoveal ILM peeling range is wide enough to include the retinoschisis area, and the foveal ILM can be left as small as possible, resulting in improvement of the macular structure. Furthermore, we could reduce the risk of postoperative macular hole formation by adding the inversion of the ILM.

Visual acuity was significantly improved at 3 months and 6 months post-surgery, with recovered or improved foveal contour in 17/20 eyes (85.0%) at 3 months and 20/20 eyes (100%) at 6 months after surgery. We consider that this was due to incomplete foveal contour recovered of the macula at 3 months after surgery. In this study, at 3 months post-surgery, 5/20 eyes (25%) showed complete recovery in foveal contour, whereas after 6 months, it had increased to 10/20 eyes (50%); by the last follow-up observation, it had increased to 18/20 eyes (90%). This indicates that foveal contour improvement of the outer layer takes a long time after surgery, especially in patients with foveal detachment.

Ho et al. reported the long-term outcome of fovea-sparing ILM for MF and concluded that the repair of the inner segment/outer segment line defect site required a prolonged postoperative course [32]. In this study, at 3 months after surgery, only 2/20 eyes (10%) were found to have EZ continuity, but by 6 months, 10/20 eyes (50%) were found to have EZ continuity, which increased to 14/20 eyes (70%) at the last follow-up observation. The foveal contour by OCT also showed improvement from 6 months after surgery to the last observation, suggesting that the foveal contour, especially the EZ, may be repaired after 6 months, improving the morphology of the retina and resulting in the improvement of visual acuity seen after 6 months post-operation.

This research, however, is subject to several limitations. First, the myopic eyes in some patients showed blurred post-surgery OCT images, even using the Spectralis Heidelberg Retina Angiograph-OCT, due to their axial lengths. The precise mechanism of vision enhancement may not be known unless all eyes are examined with a higher-resolution OCT. The second limitation is that this is a retrospective study of a small number of patients, and no controls were included. Finally, the postoperative observation period may have been too short (only 6 months) in some patients in this study.

## 5. Conclusions

This new surgical procedure, combining inverted ILM peeling and a fovea-sparing technique, was effective for treating MF without macular holes.

## Figures and Tables

**Figure 1 jcm-11-01274-f001:**
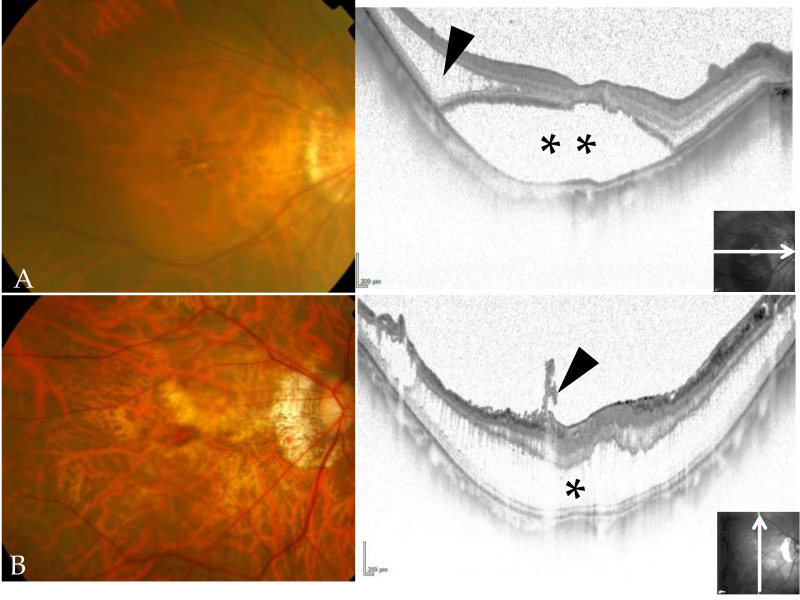
Typical color fundus photography and optical coherence tomography (OCT) images showing foveal detachment and retinoschisis. (**A**) Color fundus photograph and OCT image of a patient with high myopia showing typical myopic foveoschisis (black arrowhead) and foveal detachment without a macular hole (**). (**B**) Color fundus photograph and OCT image of a patient with high myopia showing foveal retinoschisis (*) and an epiretinal membrane, which is expected with vitreous traction (black arrowhead).

**Figure 2 jcm-11-01274-f002:**
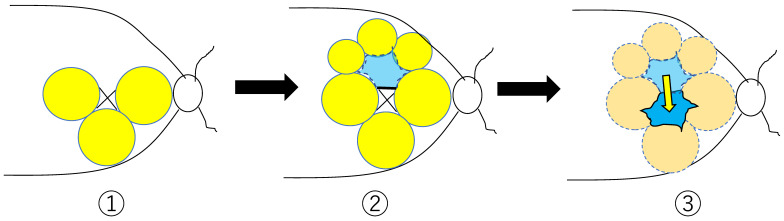
Internal limiting membrane (ILM) surgical technique. 1—Peel the ILM slightly apart from the fovea leaving only the superior fovea (yellow area). 2—Form a flap using the superior fovea (blue area). 3—Flip the flap upside down and place it on the fovea using a water stream of air-fluid exchange.

**Figure 3 jcm-11-01274-f003:**
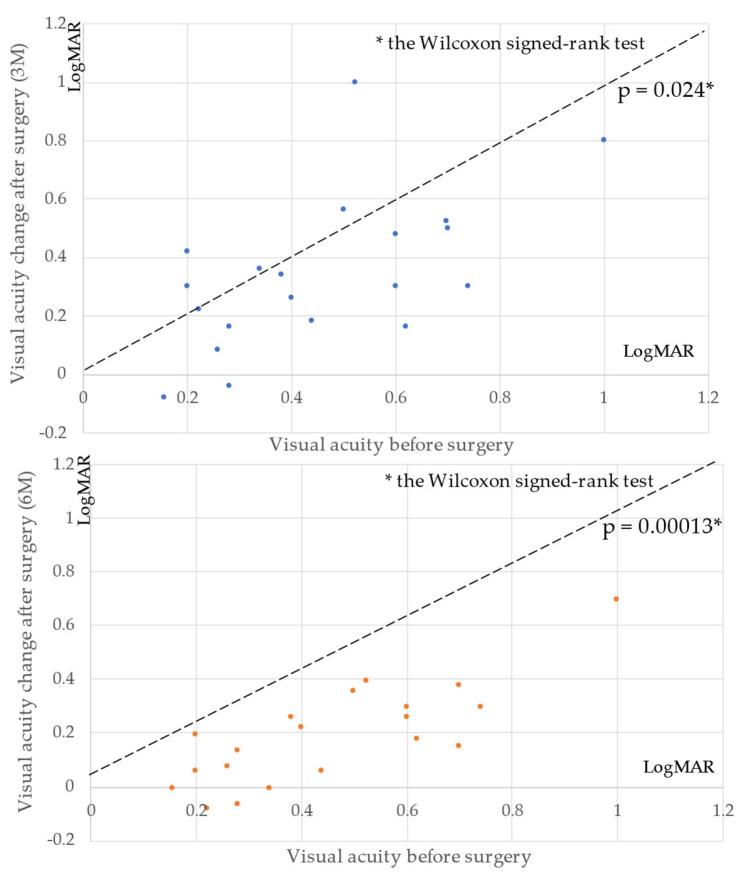
Comparison of visual acuity (best-corrected visual acuity; BCVA) before surgery with 3 and 6 months after surgery. Visual acuity showed significant improvement both 3 and 6 months postoperatively compared with preoperative acuity (*p* = 0.024 and *p* = 0.00013, respectively).

**Figure 4 jcm-11-01274-f004:**
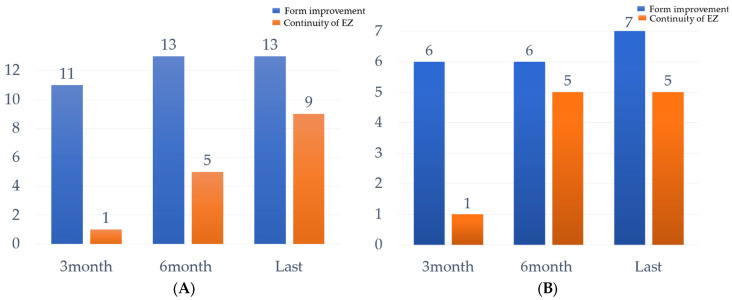
Number of patients with foveal contour improvement and continuity of the ellipsoid zone (EZ) based on OCT findings. Foveal contour improvement and EZ continuity in (**A**) patients with foveal detachment (*n* = 13) and (**B**) patients with retinoschisis (*n* = 7) at 3 and 6 months after surgery and at the last follow-up observation. In 20/20 patients, the foveal contour was improved without macular hole formation at the last observation, and EZ continuity was observed in 14/20 patients at the last observation.

**Figure 5 jcm-11-01274-f005:**
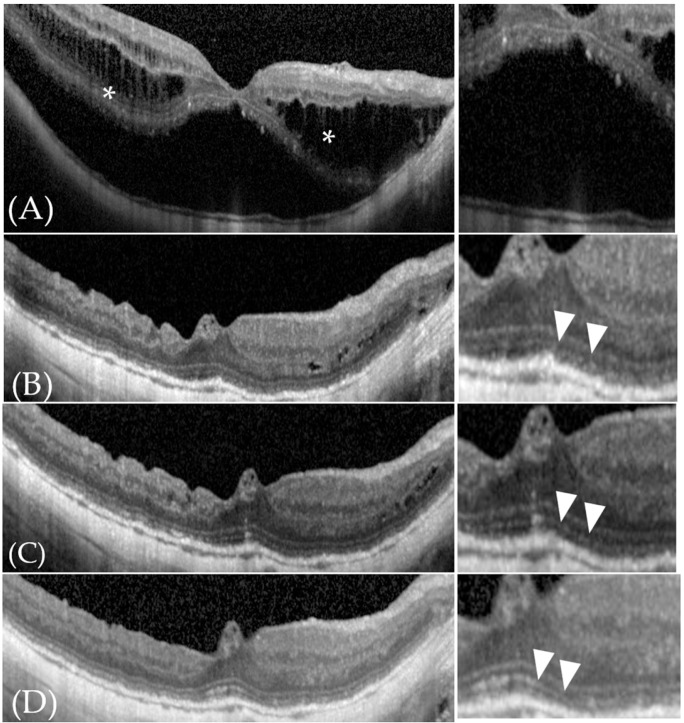
Typical OCT images of foveal detachment in a 67-year-old man (Patient 17 in Table 1). (**A**) Before surgery, we recognized retinoschisis (white asterisk) and retinal detachment of the fovea. His visual acuity was LogMAR 0.28. (**B**) Three months after surgery, the retinal detachment had disappeared and the foveal contour showed improvement without macular hole formation and so was classified as “improvement”. Although the EZ had recovered, it was obscured and discontinuous and was classified as “irregular” (white arrowheads). His visual acuity was LogMAR 0.16. (**C**) Six months after surgery, the foveal contour remained unchanged compared with 3 months after surgery and was classified as “improvement”. Although the EZ had recovered further, it was partially obscured (white arrowheads) and was classified as “irregular”. His visual acuity was LogMAR 0.14. (**D**) 1 year after surgery (last follow-up observation), the foveal contour was unchanged compared with 3 and 6 months after surgery and was classified as “improvement”. The EZ had recovered further and continuity was observed and so was classified as “continuous” (white arrowheads). His visual acuity at this stage was LogMAR 0.1. The macular hole was not recognizable after 1 year.

**Figure 6 jcm-11-01274-f006:**
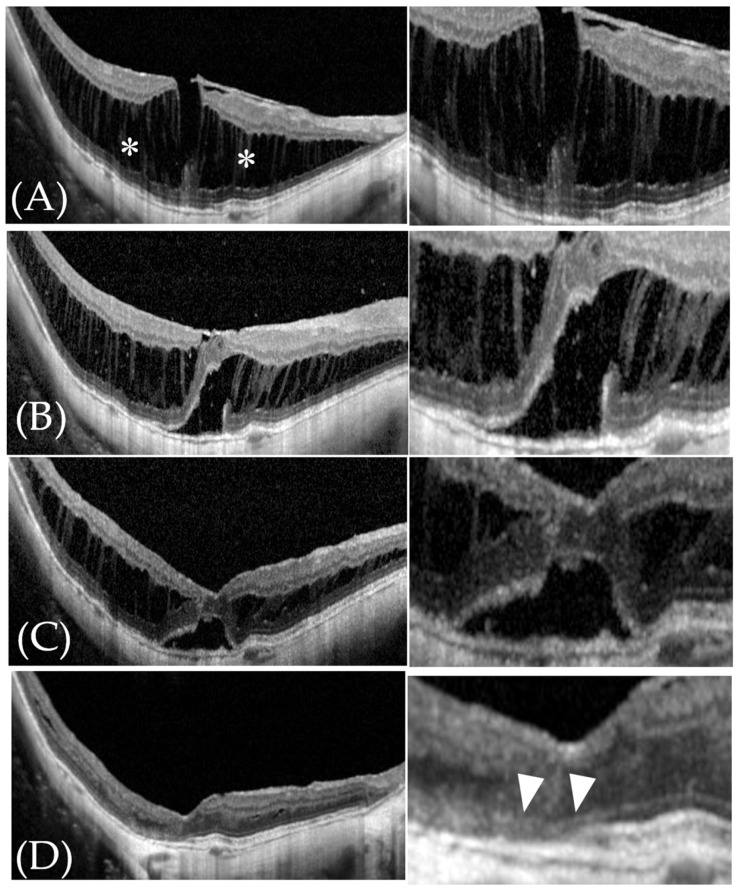
Typical OCT images of retinoschisis in a 71-year-old man (Patient #5 in Table 1). (**A**) Before surgery, we recognized retinoschisis (white asterisks) without retinal detachment. His visual acuity was LogMAR 0.5. (**B**) Three months after surgery, an outer lamellar macular hole (OLMH) was observed in this patient and the foveal contour was classified as “worse”. The EZ was discontinuous and classified as “absent”. His visual acuity was LogMAR 0.56. (**C**) Six months after surgery, both retinoschisis length and macular hole length had decreased compared with 3 months after surgery, but still remained. The foveal contour was classified as “worse”. The EZ was still discontinuous and classified as “absent”. His visual acuity was LogMAR 0.36. (**D**) One year after surgery (last follow-up observation), the OLMH and retinoschisis had disappeared and the foveal contour was improved and classified as “improvement”, but EZ discontinuity remained and was classified as “absent” (white arrowheads). His visual acuity had improved to LogMAR 0.24.

**Figure 7 jcm-11-01274-f007:**
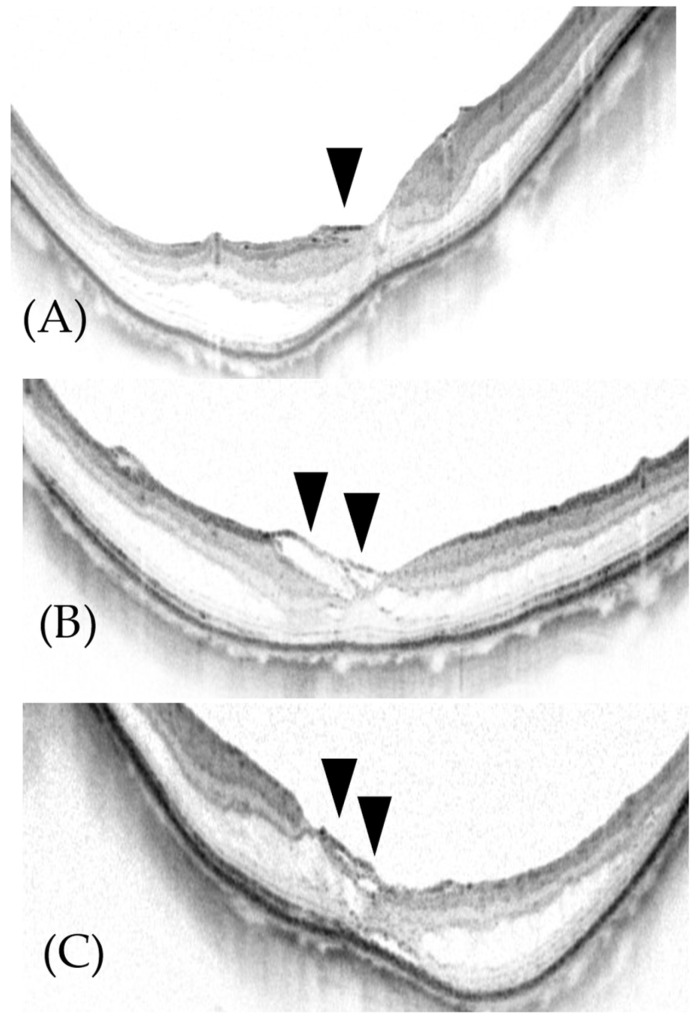
ILM flap identified in the macula by OCT. (**A**) In Patient #2 in Table 1, 3 months after surgery for retinoschisis, the ILM flap can be seen on the fovea (black arrowhead). (**B**) In Patient #11 in Table 1, 6 months after surgery for foveal detachment, the ILM flap can be seen on the fovea (black arrowheads). (**C**) In Patient #18 in Table 1, 6 months after surgery for foveal detachment, the ILM flap can be seen on the fovea (black arrowheads).

**Table 1 jcm-11-01274-t001:** Clinical characteristics, classification based on optical coherence tomography (OCT) and surgical treatment of myopic foveoschisis.

Patient No.	Age (y)	Sex	Eye	Preoperational Type by OCT	Axial Length (mm)	Tamponade	Observation Period (mo)	Operative Method
1	45	Male	Left	Retinoschisis	27.75	Air	16	Vitrectomy
2	61	Male	Right	Retinoschisis	29.81	C3F8	6	PEA + IOL, vitrectomy
3	62	Male	Left	Retinoschisis	28.64	SF6	11	PEA + IOL, vitrectomy
4	67	Male	Right	Retinoschisis	29.40	Air	11	PEA + IOL, vitrectomy
5	71	Male	Right	Retinoschisis	30.16	SF6	14	PEA + IOL, vitrectomy
6	70	Female	Left	Retinoschisis	31.10	SF6	7	PEA + IOL, vitrectomy
7	71	Female	Right	Retinoschisis	28.54	SF6	6	PEA + IOL, vitrectomy
8	70	Female	Right	Foveal detachment	28.35	Air	13	Vitrectomy
9	66	Female	Left	Foveal detachment	30.70	C3F8	26	PEA + IOL, vitrectomy
10	57	Female	Right	Foveal detachment	29.98	C3F8	68	PEA + IOL, vitrectomy
11	61	Female	Left	Foveal detachment	29.18	C3F8	93	PEA + IOL, vitrectomy
12	68	Female	Left	Foveal detachment	30.01	C3F8	19	PEA + IOL, vitrectomy
13	73	Female	Right	Foveal detachment	27.52	SF6	12	Vitrectomy
14	26	Female	Right	Foveal detachment	30.67	Air	14	Vitrectomy
15	64	Female	Left	Foveal detachment	30.20	C3F8	13	PEA + IOL, vitrectomy
16	69	Female	Left	Foveal detachment	25.59 *	SF6	6	PEA + IOL, vitrectomy
17	67	Male	Right	Foveal detachment	33.27	SF6	11	Vitrectomy
18	61	Female	Right	Foveal detachment	27.60	SF6	46	PEA + IOL, vitrectomy
19	67	Male	Left	Foveal detachment	26.50	SF6	14	PEA + IOL, vitrectomy
20	46	Female	Left	Foveal detachment	30.66	SF6	12	PEA + IOL, vitrectomy

* The axial length of the eye was less than 26 mm, but the refractive error >−6 D. The patient had a typical myopic fundus (posterior staphyloma, thin choroid, peripapillary diffuse atrophy).

**Table 2 jcm-11-01274-t002:** Foveal contour and ellipsoid zone (EZ) continuity changes following vitrectomy with Fovea-sparing and Inverted internal limiting membrane flap technique.

Patient No.	Foveal Contour (3 mo)	EZ (3 mo)	Foveal Contour (6 mo)	EZ (6 mo)	Foveal Contour (Last Observation)	EZ (Last Observation)
1	Improvement	Irregular	Improvement	Continuous	Recovered	Continuous
2	Improvement	Absent	Improvement	Continuous	Improvement	Continuous
3	Improvement	Continuous	Improvement	Continuous	Improvement	Continuous
4	Improvement	Absent	Improvement	Continuous	Recovered	Continuous
5	Worse	Absent	Worse	Absent	Recovered	Absent
6	Improvement	Recovered	Recovered	Absent	Irregular	Irregular
7	Improvement	Recovered	Recovered	Irregular	Continuous	Continuous
8	Worse	Absent	Improvement	Absent	Recovered	Continuous
9	Worse	Absent	Improvement	Absent	Recovered	Absent
10	Improvement	Absent	Recovered	Continuous	Recovered	Continuous
11	Improvement	Absent	Improvement	Absent	Recovered	Continuous
12	Recovered	Absent	Recovered	Continuous	Recovered	Continuous
13	Improvement	Absent	Improvement	Absent	Recovered	Absent
14	Improvement	Absent	Recovered	Absent	Recovered	Continuous
15	Recovered	Absent	Recovered	Absent	Recovered	Absent
16	Recovered	Absent	Recovered	Continuous	Recovered	Continuous
17	Recovered	Continuous	Recovered	Continuous	Recovered	Continuous
18	Recovered	Absent	Recovered	Absent	Recovered	Absent
19	Improvement	Absent	Improvement	Irregular	Recovered	Continuous
20	Improvement	Absent	Recovered	Continuous	Recovered	Continuous

## Data Availability

The datasets of this study are available from the corresponding author upon reasonable request.

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
