# Peer review of "Outcomes of Vitrectomy with Fovea-Sparing and Inverted ILM Flap Technique for Myopic Foveoschisis"

_jcm, 2022, doi:10.3390/jcm11051274_

Round 1
Reviewer 1 Report
Dear Author(s),
Thanks for your submission on JCM. I have read with interest the surgical procedure described in this paper and performed in 20 eyes of 20 patients with brilliant results. I encourage the authors to keep collecting the surgical cases to add more significance to the data already obtained. The OCT images provided are very explicative of some of your cases.
I have only minor concerns:
In your cohort you included the patient 16 (as in table 1), who do not have the axial length typical of an high myopic eye (>26.5 mm). Please explain.
Introduction, page 2, line 61: please delete the dash;
Page 12, line 252: please revise in "recovered";
Page 13, line 268: please revise in "post-surgery".
Author Response
Response to Reviewer 1 Comments
Thank you for such heartwarming review. We are thankful for the time and energy you expended. Responses to the referees’ comments are as follow:
Point 1: In your cohort you included the patient 16 (as in table 1), who do not have the axial length typical of an high myopic eye (>26.5 mm). Please explain.
Response 1: Thank you for pointing this out. As the Reviewer noted, the axial length of most Myopic foveoschisis eyes is longer than 26.5mm.
However, it has been reported that pathological myopia is defined as a refractive error greater than -6D or an axial length longer than 26mm. Case 16 in this study had a refractive error of more than -6D, and myopic changes in the fundus such as a posterior staphyloma, thin choroid, peripapillary diffuse atrophy, and myopic foveoschisis with detachment, so we included the case in our study even though the axial length did not exceed 26mm.
Accordingly, since the references in the Introduction were outdated, we have added the new myopia reference and changed the following text from (Introduction, page 1, line 31-33);
“MF is observed in 9%-34% of eyes with high myopia or posterior staphyloma, defined by a refractive error less than -6.0 D and an axial length greater than 26 mm [4-6]” to “MF is observed in 9%-34% of eyes with high myopia or posterior staphyloma, defined by a refractive error less than -6.0 D and/or an axial length greater than 26 mm [4-7].”
In addition, We have added a note* to the table1 (Materials and Methods, page 5, line 134-35);
“* The axial length of the eye was less than 26 mm, but the refractive error>-6 D. The patient had a typical myopic fundus(posterior staphyloma, thin choroid, peripapillary diffuse atrophy).”
Point 2: Introduction, page 2, line 61: please delete the dash;
Response 2: Thank you for pointing this out. As requested, we have delete the dash, and we have changed the following text from (Introduction, page 2, line 63):
“In this report, we devised a new technique to ma ke the treatment of MF without macular hole safer - a combination of fovea-sparing and inverted ILM flap techniques.”to
“In this report, we devised a new technique to make the treatment of MF without macular hole safe a combination of fovea-sparing and inverted ILM flap techniques. “
Point 3: Page 12, line 252: please revise in "recovered";
Response 3: Thank you for pointing this out. As requested, we have changed the following text from (Discussion, page 13, line 273-4) “We consider that this was due to incomplete foveal contour improvement of the macula at three months after surgery. “ to
“We consider that this was due to incomplete foveal contour recoverd of the macula at three months after surgery.”
Point 4:Page 13, line 268: please revise in "post-surgery".
Response 3: Thank you for pointing this out. As requested, we have changed the following text from (Discussion, page 14, line 289-91)
“First, the myopic eyes in some patients showed blurred OCT images even using the Spectralis Heidelberg Retina Angiograph-OCT, due to their axial lengths.” to
“First, the myopic eyes in some patients showed blurred post-surgery OCT images even using the Spectralis Heidelberg Retina Angiograph-OCT, due to their axial lengths.”
Reviewer 2 Report
This study demonstrates the fovea-sparing ILM peeling for retinoschisis. The manuscript is well-written and informative for retina surgeons. I have several minor comments about this manuscript.
There is a similar study in the literature. “Vitrectomy with inverted fovea-sparing internal limiting membrane for myopic foveoschisis, Journal of Ophthalmology, 2022.” The authors should show the significant contribution of their study compared with the related studies.
There are several studies introducing fovea-sparing with inverted ILM flap for macular holes and maculopathy. I think the authors should summarize these studies in one table for the literature review.
Figure 3: BCVA? UCVA?
Figure 5,6,7: Please describe the patient number.
Author Response
Response to Reviewer 2 Comments
Thank you for such heartwarming review. We are thankful for the time and energy you expended. Responses to the referees’ comments are as follow:
Point 1: I There is a similar study in the literature. “Vitrectomy with inverted fovea-sparing internal limiting membrane for myopic foveoschisis, Journal of Ophthalmology, 2022.” The authors should show the significant contribution of their study compared with the related studies.
Response 1: Thank you for pointing this out. As the Reviewer noted, we have added a note about the differences between our study and theirs. The sentences were not well connected, so I combined them with the previous paragraphs. As requested, we have changed the following text from (Discussion, page 12, line 244-52)
“post-operative OCT showed fragments of the inverted ILM flap in the superficial layer of the macular retina (Figure 7). There was a concern that this might interfere with EZ recovery or improvement of foveal contour, but we observed the morphology of the EZ in all eyes after surgery, and at last observation, EZ improvement was observed in 70% of eyes, indicating that the ILM flap does not interfere with regeneration of the EZ.” to
“Furthermore, Lin et al. said the ILM fragments could be detected by OCT in some layers on the foveal inner surface, but they have not followed the postoperative course of retinal structures such as EZ. On the other hand, we observed the morphology of the EZ in all cases after surgery, and some post-operative OCT showed fragments of the inverted ILM flap in the superficial layer of the macular retina (Figure 7). There was a concern that this might interfere with EZ recovery or improvement of foveal contour, but we observed the morphology of the EZ in all eyes after surgery, and at last observation, EZ improvement was observed in 70% of eyes, indicating that the ILM flap does not interfere with regeneration of the EZ.”
Point 2: There are several studies introducing fovea-sparing with inverted ILM flap for macular holes and maculopathy. I think the authors should summarize these studies in one table for the literature review.
Response 2: Thank you for your comments. I am sorry to say that I have not found any other paper that combines fovea-sparing and inverted ILM flap for Myopic foveoschisis or regular macular hole. However, I have found a modified version of the Inverted ILM technique to reduce the area of ILM dissection and preserve the fovea as much as possible, so we have added the new reference and changed the following text from (Introduction, page 2, line 47-50);
“Recently, a modification of the Inverted ILM technique has been reported, in which only a portion of the ILM is debrided to reduce the debrided area and minimize surgical trauma. Decreasing the area of ILM peeling has the advantage of inducing fewer changes in central retina [23,24].”
Point 3: Figure 3: BCVA? UCVA?
Response 3: Thank you for pointing this out, it is BCVA. As the Reviewer noted, our original expression here tended to be confusing. We have added a note to the Figure3 and changed the following text from (Results, page 6, line 141-42);
“Figure 3. Comparison of visual acuity before surgery with three and six months after surgery.” to
“Figure 3. Comparison of visual acuity (best-corrected visual acuity; BCVA) before surgery with three and six months after surgery.”
Point 4: Figure 5,6,7: Please describe the patient number.
Response 4: Thank you for pointing this out. As requested, we have added the patient number and changed the following text from (Figure5 and 6; Results, page9 and 10, line 165 and 182, Figure7; Discussion, page13, line 254-58) .
“Figure 5. Typical OCT images of foveal detachment in a 67-year-old man.” to
“Figure 5. Typical OCT images of foveal detachment in a 67-year-old man (Patient 17 in Table1). ““Figure 6. Typical OCT images of retinoschisis in a 71-year-old man” to
“Figure 6. Typical OCT images of retinoschisis in a 71-year-old man (Patient 5 in Table1).”
“Figure 7. ILM flap identified in the macula by OCT. (a)Three months after surgery for retinoschisis, the ILM flap can be seen on the fovea (black arrowhead). (b)Six months after surgery for foveal detachment, the ILM flap can be seen on the fovea (black arrowheads). (c) Six months after surgery for foveal detachment, the ILM flap can be seen on the fovea (black arrowheads).” to
“Figure 7. ILM flap identified in the macula by OCT. (a) In patient 2 in Table1, three months after surgery for retinoschisis, the ILM flap can be seen on the fovea (black arrowhead). (b) In patient 11 in Table1, six months after surgery for foveal detachment, the ILM flap can be seen on the fovea (black arrowheads). (c) In patient 18 in Table1, six months after surgery for foveal detachment, the ILM flap can be seen on the fovea (black arrowheads).”